# How Public Trust in Health Care Can Shape Patient Overconsumption in Health Systems? The Missing Links

**DOI:** 10.3390/ijerph18083860

**Published:** 2021-04-07

**Authors:** Katarzyna Krot, Iga Rudawska

**Affiliations:** 1Faculty of Engineering Management, Bialystok University of Technology, 15-351 Bialystok, Poland; 2Institute of Economics and Finance, University of Szczecin, 71-101 Szczecin, Poland; iga.rudawska@usz.edu.pl

**Keywords:** public trust in health care, overconsumption of health care, health system, structural equation modeling

## Abstract

Overconsumption of health care is an ever-present and complex problem in health systems. It is especially significant in countries in transition that assign relatively small budgets to health care. In these circumstances, trust in the health system and its institutions is of utmost importance. Many researchers have studied interpersonal trust. Relatively less attention, however, has been paid to public trust in health systems and its impact on overconsumption. Therefore, this paper seeks to identify and examine the link between public trust and the moral hazard experienced by the patient with regard to health care consumption. Moreover, it explores the mediating role of patient satisfaction and patient non-adherence. For these purposes, quantitative research was conducted based on a representative sample of patients in Poland. Interesting findings were made on the issues examined. Patients were shown not to overconsume health care if they trusted the system and were satisfied with their doctor-patient relationship. On the other hand, nonadherence to medical recommendations was shown to increase overuse of medical services. The present study contributes to the existing knowledge by identifying phenomena on the macro (public trust in health care) and micro (patient satisfaction and non-adherence) scales that modify patient behavior with regard to health care consumption. Our results also provide valuable knowledge for health system policymakers. They can be of benefit in developing communication plans at different levels of local government.

## 1. Introduction

Health is commonly perceived as one of the most desirable values. It can be treated as an asset forming individual capabilities, or a critical pillar of societal and economic development. Consequently, health systems play a crucial role in shaping the potential of any society [1] As the nature of health systems is a relational one, trust is centrally placed in shaping patients’ attitudes to health and their health-related behaviors [2]. Thus, to understand the behavioral patterns of health care consumers, it is important the that elements (variables) should be revealed which may affect the doctor-patient interaction and impact on the level of consumption of health care services. Both under-consumption and overconsumption may lead to negative outcomes for the whole health system, thus disrupting the attainment of its goals. The greatest challenge is in the economic burden that may manifest itself in higher medical costs per patient, inefficient resource allocation, and an ineffective patient eligibility verification process.

The nature of patient behavior in health care with the central role of trust has long been given considerable attention by researchers [3,4]. The concept of trust has been explored extensively in developed countries, mostly focusing on the United States [5], and Australia [6]. There has been much less interest in the nature of trust and its impact on health care consumption in the context of health systems of developing countries [7,8]. Moreover, while interpersonal trust and its impact on the effectiveness of the doctor-patient interaction has received a substantial focus [4,9], relatively little research has been conducted regarding public trust and its consequences [10,11,12].

This paper attempts to address the issue of patient moral hazard in health care. Specifically, the impact of public trust on shaping the behavior of patients as health care consumers is studied in order to identify and examine the link between public trust and overconsumption of health care. Moreover, the mediating role of patient satisfaction and patient non-adherence is explored.

The following structure has been adopted for the paper: first, the existing literature is reviewed in the search for a possible research gap that would warrant undertaking the proposed study. Subsequently, the research hypotheses concerning the relationships between public trust/ mistrust and the level of consumption of health care services by patients are evaluated. Finally, the concepts of patient satisfaction and patient nonadherence as mediators in shaping overconsumption of health care are introduced. As of yet, the knowledge of the effects of public trust on the level of health care consumption and moral hazard in countries in transition, such as Poland, is incomplete.

## 2. Literature Review

### 2.1. Overconsumption of Health Care and Patient Moral Hazard

There have been numerous attempts to estimate the scale of overuse of medical care, although they have not always brought the expected results. The limitations of the initial efforts mainly resulted from the difficulties in defining the concepts and, consequently, developing a methodological approach. Chassin and Galvin [13] (1998) define overuse as “the provision of medical services for which the potential for harm exceeds the potential for benefit”. The mere measurement of the benefits and harms of using medical services is a controversial issue, mainly due to incomplete data and a missing typology of services [14,15]. 

One cautious estimate made in the U.S. and based on direct measurements across individual services shows that overuse rates are between 6% and 8% of total health-care expenditure [16]. Other studies indicate that overuse losses in the U.S. are as high as 29% [17]. There have been reports of even higher percentages, although those results are only subject to limited comparison. Although overuse is a global problem, most research concentrates on high-income countries, with negligibly little interest in low and middle-income countries [15]. Brownlee et al. [15] present a number of doubts concerning both the direct and indirect methods for measuring overconsumption. At the same time, they state that such research is necessary because overuse may lead to a physical, mental, and financial harm to patients and contribute to the excessive use of the resources of the health and social care systems in both high-income and low- and middle-income countries. They point to the global scope of the phenomenon covering most medical specializations (hence, overtreatment, over-testing, over-diagnosis, overuse of medication, overuse of screening tests, etc.). Other authors point out that controlling overuse of health care may have positive impact on cost reduction and quality improvements at the same time by sparing patients the unnecessary risk that attends to inappropriate health care [13].

One special type of overuse referred to as moral hazard. It is commonly understood as an excessive expenditure due to eligibility for insurance benefits [18]. According to economists, moral hazard is perceived to be any misallocation of resources that occurs when risks are insured with only normal insurance contracts. More specifically, it is defined as health insurance bearers’ inclination to change their behavior in a way that increases the risk of loss for the insurer. According to Zweifel and Manning [19], there are two categories of moral hazard behaviors in the context of health insurance: ex ante behaviors, where the risk of loss grows prior to a medical event with individuals engaging in higher-risk activities, thus increasing the likelihood of an event causing a loss, and ex post behaviors with individuals using increasing levels of health care following the event [20]. Folland et al. [21] has found that moral hazard refers to the increasing use of services, while the marginal costs of medical services are declining.

Overconsumption of medical services that are covered by insurance, either public or purchased privately, can lead to multiple adverse effects, including an increase in the costs of medical services and a reduced availability of services for patients in actual need. In an effort to limit the overall costs of overuse of medical services, factors with the potential to help constrain such patient behavior must be identified. Research into overuse reported in the literature primarily makes use of macroeconomic analyses and focuses on attempts at estimating its size [22]. Other authors also seek to determine the relationship between co-payment schemes and the level of demand for medical services. The results reported confirm that moral hazard can be controlled through demand management, e.g., through the introduction of co-payment systems [23,24].

Additionally, Blomqvist [25] notes that, while co-payment reduces moral hazard by increasing the efficiency of health care use, it may limit the consumption of medical services, which in turn may contribute to an increased risk of disease and poorer patient welfare. Clearly, however, little research has been devoted to identifying determinants that either support or inhibit excessive consumption of medical services. For certain, though, such factors typically include the price and the type of the potentially overused medical service [26]. The studies carried out for the U.S. by Konetzka et al. [27] indicate that elderly patients overuse home care if it is funded by insurance. Zweifel and Manning [19] have attempted to identify the stimuli (on the part of patients) that may significantly contribute to the likelihood of moral hazard, among them, the patients’ age. Moreover, other studies indicate that patients’ previous experiences affect the chances of developing a tendency to overconsume medical services. At the same time, research performed in the USA shows that the overuse of hospital stays is much more likely than, for example, drug overconsumption [28]. There is definitely a need for further studies that would expand our knowledge of the factors that determine the overuse of medical services.

### 2.2. Patient Satisfaction and Patient Adherence

The health system is a professional service sector where the concept of customer satisfaction is also applicable. The most common understanding of this concept refers to a subjective evaluation of services received by a customer [29]. Patient satisfaction with the medical services has been defined a patients’ overall evaluation of the performance of a service offering after experiencing it [30]. It can be treated as an essential evaluating tool for health care delivery [31], as well as an important quality indicator of health care [32]. Patient satisfaction reflects the gap between the expected level of health care and the actual experience of the medical encounter, perceived by the patient. 

It has been proved that patient satisfaction with past outcomes positively impact social/general trust in physicians as a professional community [33]. Moreover, many authors point at a significant positive relation between overall satisfaction with the health care and overall communication behaviors [34,35], but there is no extensive evidence how these attributes of the patient-doctor relationship impact patient consumption patterns in health systems. 

World Health Organization interprets patient adherence as the extent to which a person’s behavior (such as taking medication, modifying lifestyle, following a diet) corresponds with agreed recommendations from a health care professional [36]. In the literature this term as used as an alternative to compliance [37]. Non-adherence or noncompliance is, thus, perceived as a negative behavior of a person facing a medical problem. Non-adherence to drug therapy or general to medical treatment seems to be a common problem in health systems. According to different sources, the adherence rate for long-term medication ranged from 33% to 94%, and the compliance rate in long-term therapy tend to reach at the most 50%, no matter the setting or illness [38]. This phenomenon has been recorded both in developed, as well as in developing, countries. Poor compliance to medical treatment has a negative impact on health indicators, such as morbidity and mortality rates [36]. It can be also associated with lower treatment efficacy, higher hospitalization rates, and higher health care costs [39].

It has been investigated that patients’ motivation to adhere to medical treatment is positively related to their understanding of the health condition and the treatment itself, together with their faith in the physician [40]. It can, therefore, be assumed that the non-adherence can be associated with poor understanding and beliefs about the treatment and with weak trust in a doctor. The first correlation have been shown in previous studies [41], but the second one still needs more recognition. 

### 2.3. Trust in Health Care

Trust, with its definitions attempted by different disciplines ranging from psychology to management, is an extraordinarily complex construct. It is perceived as an important feature of any relationship, both on a micro- and macroscale. The concept of trust has become especially meaningful in the service sectors of the economy that are purely focused on relational links, such as health care. 

Trust can be viewed from multiple perspectives. In social sciences, trust is understood as an infinite process [42] and the foundation of social interactions [43]. It is believed to be a valuable resource for systems, organizations and interpersonal relationships [44]. According to Hardin [45], trust is an alternative to credibility which, ensured by various social solutions, institutions, or standards, allows one to take actions based on confidence. Putman [46] believes that trust is an element of social capital that “*refers to features of a social organization such as trust, norms and networks that can improve the efficiency of society by facilitating coordinated actions*”. In economics, trust is perceived as an “informal norm” that reduces the costs of transactions that supervise the conclusion of contracts, the enforcement of formal agreements and the settlement of disputes [47]. According to the behavioral approach prevailing in psychology, trust typically means “holding a positive perception about the actions of an individual or an organization” [48]. It is commonly described as the willingness to rely on a partner in whom one has confidence [49]. Trust allows us to believe that others will behave as they are expected to, whether on a one-off basis or in a set of situations. Although trust may be developed through actual experience, it is often subjective and based as much on interpretation as on facts [48].

Any discussion about trust is hindered by the wide range of available classifications of this relational aspect. Comprehensive reviews of these classifications have been prepared by Jabłoński and Jabłoński [50] and OECD [48]. In the present paper, the focus is only on those that can be applied to health care. And so, as regards trust in medical professionals there are two closely related concepts identified: general trust people have in physicians as a whole, on the one hand; and interpersonal trust found in a relationship with the particular physician, on the other [51]. Theoretically, patients’ general trust in physicians contributes to inducing their interpersonal trust in a newly-acquainted doctor and to maintaining that trust as the interactions develop [52]. Based on the social theories of Giddens and Luhmann, it has been shown that these two types of trust are engaged in an interaction resembling a complex helix [53]. Studies have shown that general trust in doctors correlates with trust in the particular physician, thus encouraging satisfaction with the care received, better medical adherence [51,54] and fulfillment of medical needs [9]. Moreover, interpersonal trust may contribute to the patients introducing and maintaining a self-care management regime, and adhering to treatment [55,56]. It has also been proven empirically that higher levels of trust in medical professionals have a positive impact on patient outcomes, e.g., in such areas as glycemic or blood pressure control [57], which may entail lower health-care costs [3]. In turn, a patient’s mistrust leads to them adopting undesirable behaviors and attitudes, resulting in poor satisfaction with the care received [58] and a reduced degree of interaction with the health system [59]. Patients who do not follow medical advice and fail to adhere may need extra treatment in the future, which translates into medical expenditure escalation on the macroscale. Thus, mistrust can potentially increase medical expenditure and lead to inefficient resource allocation [60].

The literature also features analyses of the concept of public trust. This form of trust is one that a person or a group of persons place in a system or a societal institution, such as a health system [10]. Similarly to general trust in medical professionals, public trust in a health system and an individual’s interpersonal trust may be mutually supportive [7]. Public trust can determine how patients develop their interactions with health care providers, whereas the experience they derive from contacts with health care institutions and their representatives can in turn impact on public trust. Public trust can be, thus, associated with how patients and the general public perceive the health system and its ability to deliver services and, eventually, to meet its goals [1]. Moreover, it can provide a significant indicator of the level of support for the health system as a whole [1]. Public trust, also referred to as social trust, informs interpersonal trust [61]. It can be influenced not only by the past experiences of a particular patient or their generalized social confidence in the public institutions but also by the experiences of others and by the mass media [62]. 

Public trust is of critical importance in a service encounter, especially in the case of interactive services, such as health care. It determines possible future behaviors of patients and, therefore, can be created as a key asset of the medical profession and the health system as a whole. Based on the findings reported in the literature of the English-speaking world that relate mostly to developed countries [63,64] or to the race minorities found within them [65,66], public trust can be claimed to impact on an array of health behaviors. It can be assumed that higher trust levels are typically associated with positive behaviors. In the health care context, it may be assumed that public trust creates a higher level of patient satisfaction and leads to better compliance with medical treatment, which in turn ultimately influences the level of consumption of health-care services [67]. On the other hand, a lack of trust (mistrust) entails poor clinical relationships that exhibit less continuity, poorer adherence to the doctors’ recommendations, and a reduced degree of patient-physician interaction [66]. 

Public trust is a multidimensional concept. Within the research context, it is usually explored through several dimensions that relate to prioritizing patients’ interests, care quality, the service providers’ expertise, a patient-centered focus of health care, the effects of health system policies, access to health-care products and services, and information and communication provision [1,63,64,68,69]. These dimensions reflect the diverse considerations on how public trust should be understood and measured, going far beyond the technical competences of medical professionals. These aspects have also been accommodated within the scale designed as part of our study to measure public trust. 

While interpersonal trust has been studied extensively, social trust in the health system, especially for countries in transition, has gained far less scientific attention. The existing literature presents findings concerning predictors of trust in health systems [70], antecedents of patient trust in health-care insurance [71], general trust in the medical profession [51], fiduciary trust in physicians and its outcomes [72]. As there is limited research on public trust and health care use in the systems in transition, it is currently unclear how these two constructs interrelate and what the roles of patient satisfaction and medical noncompliance are. In our study we are going to fill in this gap. Therefore, this study is specifically concerned with the concepts of patient satisfaction and patient non-adherence as mediators in shaping overconsumption in health systems. Our work is based on the key terms discussed in Section 2.1, Section 2.2 and Section 2.3. Table 1 shows a brief summary of the presented concepts.

## 3. Materials and Methods

### 3.1. Research Model and Hypotheses

Patients with high public trust in health care may be less likely to overuse the health care resources than health system users demonstrating low public trust. Moreover, they may be more motivated to adhere to medical regime and more satisfied with the medical services, which, in turn, may manifest itself in avoiding unnecessary visits. This reasoning leads us to the following hypothesis:

**Hypothesis** **1** **(H1).**
*Public trust in the health system correlates negatively with overconsumption of health care by patients.*


We assume that overuse of health care may be the result of both macro-level phenomena (such as trust in the system) and interpersonal relations (such as patient satisfaction). Moreover, we propose that one’s approach to adhering to medical recommendations may have an impact on the extent of health care use. Patients demonstrating carelessness by arbitrarily changing their doctor’s recommendations will also be more likely to overuse medical care. Figure 1 shows the above assumptions in the form of a model explaining the relationships between the variables. 

Additionally to H1, we propose five more hypotheses concerning the relationship between public trust and the referents of patient-physician relationships (patient satisfaction; patient noncompliance/non-adherence), and their links to overconsumption of health care by patients:

**Hypothesis** **2** **(H2).**
*Public trust in the health system correlates positively with patient satisfaction.*


**Hypothesis** **3** **(H3).**
*Public trust in the health system correlates negatively with patient nonadherence.*


**Hypothesis** **4** **(H4).**
*Patient satisfaction with the doctor correlates negatively with patient nonadherence.*


**Hypothesis** **5** **(H5).**
*Patient satisfaction with the doctor correlates negatively with overconsumption of health care by patients.*


**Hypothesis** **6** **(H6).**
*Patient non-adherence correlates positively with overconsumption of health care by patients.*


### 3.2. Variables and Scale Development

The scales for measuring individual variables were designed based on literature reports (Table 2). The constructs and items were adopted from the sources indicated in Table 2. Each of the statements was proposed to the respondents along with a five-point Likert scale.

Trust in the system is a construct measured with the use of 5 items relating to service quality, medical care standards, staff competence, and free access to doctors and medicines (e.g., *“Medical services are of high quality”, “I believe that health-care institutions try to employ the best doctors”, “Patients have free access to the doctors they prefer*”). The Alpha Cronbach value for this scale assumed a value of 0.68.

Patient satisfaction is a variable that is often used in studies and has, therefore, been repeatedly verified. It has been measured by the following statements, among others: *“Most often I am satisfied with the doctor’s attitude towards me (respect, sensitivity of interest)”, “I am satisfied with the way the doctor talked to me”, “I am satisfied with the doctor’s level of knowledge”*. The Alpha Cronbach value for this scale was 0.86. 

Patient non-adherence, in turn, was examined using the following three items: “It happens to me that I do not take my prescription to the pharmacist”, “It happens to me that I stop taking the medicine without informing the doctor”, and “If I feel better, it happens to me that I give up treatment”. The Alpha Cronbach value for this scale was 0.64. This variable was identified as a factor in the factor analysis of the medical adherence scale. There were several reasons for only taking into account one of the model’s dimensions, i.e., patient non-adherence. First of all, as in the case of moral hazard, this particular patient behavior was deemed as unfavorable, both from the perspective of the individual and in the context of the entire health system. Moreover, it was assumed that patients who are predisposed to a “voluntary” approach to the doctor’s recommendations will also be more prone to overconsumption of medical services. Thus, the occurrence of patient non-adherence may reinforce overconsumption of health care by patients.

The study attempted to examine only one type of patient moral hazard situations, i.e., ex post behaviors; therefore, the research model assumed that the dependent variable would be “overconsumption of health care by patients”. This variable was tested with the use of one statement, namely: “*I believe that in some cases it is necessary to visit several (two, three or even four) doctors of the same specialty with the same ailment to confirm the diagnosis*”. The adopted measurement method served rather to confirm the occurrence of this phenomenon than to estimate its size.

### 3.3. Sample and Method

The study, based on the prepared questionnaire, was carried out by the international research agency Millward Brown. The representative sample was selected at random. The study was carried out in 2015 and 2016 using the CATI method on a nationally representative sample of 982 Polish respondents declaring having used health-care services within the most recent 6 months. Table 3 presents characteristics of the study sample.

Data analysis and model verification was performed using SPSS and Amos software of IBM Corporation (Somers, NY, USA).

## 4. Results

The descriptive statistics of the variables are presented in Table 4. It is worth noting that some of the correlations are negative.

To verify the theoretical model, structural equation modeling (SEM) was used, a linear cross-sectional statistical modeling technique that includes path analysis and regression analysis. SEM is used to explain the pattern in a series of interrelated dependence relationships simultaneously between a set of latent constructs. In addition, the technique is also used to estimate variance and covariance, test hypotheses, conventional linear regression, and factor analysis [75]. However, SEM-tested model must be based on theoretical assumptions and only theory can stimulate and trigger the development or modification of the model. Since SEM is mostly used to determine whether a certain model is valid rather than to “find” a suitable model, it is the most applicable statistical method to validate the proposed model (Figure 1). This is where the theory plays an important role in justifying the model [76].

To verify the theoretical model, the maximum likelihood (ML) estimation method was applied. The ML function is a structured means model reflecting how closely the sample mean vector is reproduced by the estimated model mean vector. It also indicates how closely the sample covariance matrix is reproduced by the estimated model covariance matrix [77].

The results revealed a χ^2^ of 399.78 based on 224 degrees of freedom with a probability level of 0.00. As the indicators show, the goodness-of-fit measures of the model were satisfactory (Table 5).

All the model paths were statistically significant, which means that all the hypotheses were confirmed (Figure 2). This indicates that all the elements on the left side of the model affected the extent of overconsumption of medical services. In addition, the direct impact of the studied phenomena also encompassed their indirect impact (Table 6).

Failure to comply with the doctors’ recommendations, as manifested in the patients arbitrarily modifying the prescribed therapy (H6), had the most substantial impact on overconsumption of health care by patients. Apart from this, overconsumption depended on the nature of the relationship with the doctor (H5) and on public trust in the health system (H1). Both patient satisfaction (the interpersonal level) and public trust in the system (the macroscale level) were statistically significant barriers discouraging patients from unjustified use of medical services (a negative impact). Importantly, the strength of the influence of both the factors was comparable.

Interestingly, public trust in the system was shown to be an overarching construct that definitely had a strong strengthening effect on patient satisfaction in a relationship with the doctor (H2). At the same time, this trust prevented patients from arbitrarily modifying their doctors’ recommendations (H3). Hypothesis 4 was also confirmed, suggesting that in the patient-doctor relationship patient satisfaction played a significant role in limiting the patients’ failure to comply with medical recommendations. 

The results of the study also proved that public trust in the health system constituted a special construct in the model being described herein, since it influenced both directly and indirectly how medical services were used (Table 7). The negative impact of public trust in the health system on overconsumption of health care by patients was reinforced by patient satisfaction. This means that the coexistence of both these factors at different levels of the system, i.e., trust on a macroscale and patient satisfaction on a microscale, were definitely more effective at preventing patients from overusing medical services. Importantly, public trust in the health system, thanks to coexisting with patient satisfaction, also significantly reduced patient non-adherence.

## 5. Discussion

Overconsumption of health care by patients is an undesirable phenomenon, both from the perspective of the entire system and the doctor-patient relationship. Previous research shows that overconsumption of medical services primarily leads to an increase in treatment expenditure, inefficient resource allocation, and an ineffective eligibility process [19,36,39]. It is, therefore, legitimate to look for strategies to reduce this trend and its negative effects. In the literature, patient moral hazard has been studied mainly from an economic perspective, with various attempts at estimating the degree of overconsumption of medical services [19,21,22]. In the present study, completely different optics were adopted. Instead of being estimated, overuse of health care was rather treated as a dependent variable. Therefore, it was identified as the patients’ declared use of the same category of medical services more than once in order to confirm an earlier diagnosis. Thus, the aim of the study was not to focus on the degree of overconsumption but to explore the factors influencing the occurrence of overuse. Few studies have thus far focused on identifying factors contributing to or preventing overconsumption of medical services [19]. Our research sought to expand knowledge in this particular area.

Our research shows that overconsumption of health care by patients is most strongly inhibited if patients have trust in the health system, whereas the strength of that correlation is subject to either direct (H1) or indirect (patient satisfaction) impact. The patients’ belief that the health system is well managed assures them of the high quality of the medical care they will receive. Moreover, their belief that the doctors are competent prevents them from reverifying their diagnoses. These results are consistent with the outcomes of the research conducted by Montague et al. [62] and Kato and O-Malley [60]. Moreover, the significant moderating role of patient satisfaction must be emphasized. Trust that patients have in the entire health system strongly determines how they feel in the personal relationship with their doctor. The more the patient trusts the system, the more satisfying their relationship with their doctor is (H2). Consequently, both these factors—namely, trust in the health system and patient satisfaction (H2)—constitute significant barriers to overconsumption of medical services. Additionally, patient satisfaction reduces patient non-adherence, which also weakens overconsumption. Accordingly, patients satisfied with medical care demonstrate more “ethical” behavior, where they and the entire system are not exposed to significant and unpredictable expenditure. A patient who trusts the health system and is satisfied with the medical services received will be more likely to use the limited available resources fairly, leaving some for others in need of care to benefit from.

The conceptual model for this research argues that public trust influences the level of health care consumption and that patient satisfaction and patient non-adherence play a mediating role in the process. Especially, research findings on non-adherence as an independent factor are relatively rare.

The findings presented herein add to the existing knowledge of the links between public trust and overconsumption in health systems. This study provides a number of contributions:

It is an attempt at taking a new and two-dimensional look at overconsumption of health care. On the one hand, overuse is a macroeconomic problem, and it has been the subject of extensive research. On the other hand, it can be viewed as patient behavior that is subject to modification through appropriate management by the service provider (patient satisfaction) and by the health system (public trust in health care).It identifies factors influencing overconsumption of health care, both on the macro (trust in the health system) and micro (patient satisfaction, non-adherence) scales. It also points to the mediating role of patient satisfaction and emphasizes the importance of patient non-adherence in stimulating overuse of medical services.It focuses on Poland, a country in transition, where no research into the degree of overuse has been conducted before. Interestingly, Poland’s health system has been undergoing major transformations, and patients are not yet treated as clients.Finally, it improves the understanding of how public trust operates within both macro- and micro-relationships in the health system, which may provide an insight into interpersonal trust. This research is sufficiently novel in that few studies have emphasized the role of patient satisfaction and patient non-adherence in shaping overconsumption of health care before.

This study encountered certain limitations that may have affected its outcome. First of all, the method used to study overconsumption may be regarded as controversial. The approach adopted was a consequence of the study’s objectives, although that choice is not popular in the literature. In further similar research, the scale for measuring overuse of health care by patients should, therefore, be refined and verified. Another limitation is related to the selection of explanatory variables used in the research model. Any further research should expand the scope to include other dimensions of compliance with medical recommendations and, perhaps, trust in doctors.

## 6. Conclusions

This study of overuse of health care by patients is not exhaustive, as gaps in the collective knowledge of the antecedents of this particular construct remain. Hopefully, some of the gaps were made narrower by the findings made herein. Results show that public trust in health care is associated with improved chances of shaping the consumption patterns among patients. Increasing patients’ public trust may, thus, play a significant role in reducing health care costs.

## Figures and Tables

**Figure 1 ijerph-18-03860-f001:**
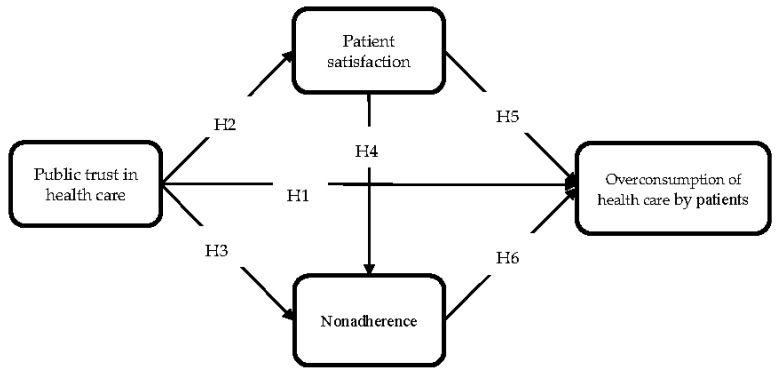
A conceptual research model.

**Figure 2 ijerph-18-03860-f002:**
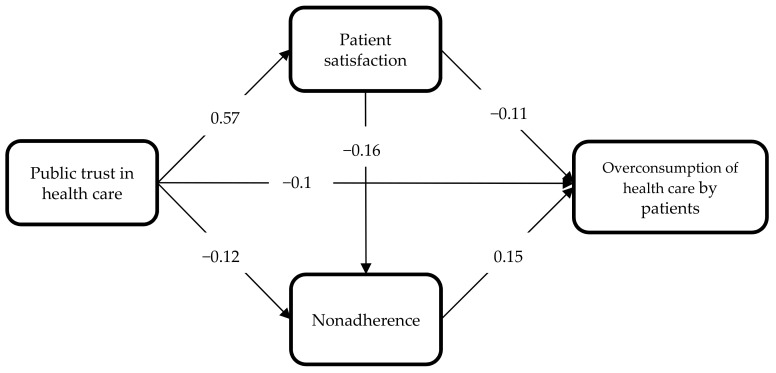
Model of relationship between public trust in health care and overconsumption of health care. Source: developed by the authors.

**Table 1 ijerph-18-03860-t001:** The key concepts of the study.

Concept	Meaning	Source
Overuse	A situation when health care service is provided under circumstances in which its potential for harm exceeds the possible benefit	[13]
Moral hazard	Excessive expenditure due to eligibility for insurance benefits	[18]
Patient satisfaction	A patients’ overall evaluation of the performance of a service offering after experiencing it	[30]
Patient adherence	The extent to which a person’s behavior (such as taking medication, modifying lifestyle, following a diet) corresponds with agreed recommendations from a health care professional	[36]
Interpersonal trust	Characterizes a relationship between two individuals, such as a specific doctor–patient relationship.	[51]
Public/system trust	Characterizes attitudes toward collective entities or social organizations.	[10]

Source: developed by the authors.

**Table 2 ijerph-18-03860-t002:** Scale characteristics.

Variable	Source	Cronbach Alpha
Public trust in health care (5 items)	[63,64]	0.68
Patient satisfaction (8 items)	[64,73,74]	0.86
Patient non-adherence (3 itmes)	[39]	0.64

Source: developed by the authors.

**Table 3 ijerph-18-03860-t003:** Study sample structure.

Income			Sex		
	N	%		N.	%
up to PLN 1.000 *	84	9.3	female	572	58.2
from PLN 1.001 to 1.400	95	10.5	male	411	41.8
from PLN 1.401 to 1.800	108	12.0	Age		
from PLN 1.801 to 2.000	113	12.5	18–24	103	10.5
from PLN 2.001 to 2.500	76	8.4	25–34	182	18.5
from PLN 2.501 to 3.000	120	13.3	35–44	163	16.6
from PLN 3.001 to 5.000	139	15.4	45–59	252	25.6
over PLN 5.000	118	13.1	over 60	283	28.8
hard to say	49	5.4			
Place of residence			Level of education		
village	356	36.2	elementary	148	15.0
up to 100 thous.	335	34.1	basic vocational	213	21.7
100–499 thous.	164	16.7	secondary	345	35.2
500+ thous.	127	13.0	college/university	276	28.1

* 1PLN = 0.2167 EURO according to the average exchange rate of the National Bank of Poland as at 23.03.2021. Source: developed by the authors.

**Table 4 ijerph-18-03860-t004:** Descriptive statistics (mean, standard deviation, and correlation coefficient).

	Mean	SD	Patient Satisfaction	Patient Non-Adherence	Public Trust in Health Care
			Correlation Coefficient		
Patient satisfaction	3.62	0.77			
Patient non-adherence	2.31	1.02	−0.19		
Public trust in health care	2.97	0.76	0.47	−0.14	
Overconsumption of health care by patients	3.54	1.22	−0.19	0.16	−0.15

Source: developed by the authors.

**Table 5 ijerph-18-03860-t005:** The goodness of fit indexes.

Index	Score
CMIN/DF	1.78
GFI	0.97
CFI	0.97
RMSA	0.028
Holter	691

Source: developed by the authors.

**Table 6 ijerph-18-03860-t006:** Standardized regression weights.

			Estimate	*p* Value	
H1: Overconsumption of health care by patients	←	Public trust in health system	−0.10	0.002	Supported
H2: Patient satisfaction	←	Public trust in health system	0.57	0.000	Supported
H3: Patient non-adherence	←	Public trust in health system	−0.12	0.05	Supported
H4: Patient non-adherence	←	Patient satisfaction	−0.16	0.003	Supported
H5: Overconsumption of health care by patients	←	Patient satisfaction	−0.11	0.000	Supported
H6: Overconsumption of health care by patients	←	Patient non-adherence	0.15	0.001	Supported

Source: developed by the authors.

**Table 7 ijerph-18-03860-t007:** Standardized total, indirect, and direct effects.

	Public Trust in Health Care	Patient Satisfaction	Patient Non-Adherence
Standardized total effects			
Public trust in health care	-	-	-
Patient satisfaction	0.57	-	-
Patient non-adherence	−0.21	−0.16	-
Overconsumption of health care by patients	−0.19	−0.13	0.14
Standardized direct effects			
Public trust in health care	-	-	-
Patient satisfaction	0.57	-	-
Patient non-adherence	−0.12	−0.16	-
Overconsumption of health care by patients	−0.10	−0.11	0.14
Standardized indirect effects			
Public trust in health care	-	-	-
Patient satisfaction	-	-	-
Patient non-adherence	−0.09	-	-
Overconsumption of health care by patients	−0.09	−0.02	-

Source: developed by the authors.

## Data Availability

Data is contained within the article.

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
