# Peer review of "How Public Trust in Health Care Can Shape Patient Overconsumption in Health Systems? The Missing Links"

_ijerph, 2021, doi:10.3390/ijerph18083860_

Round 1

Reviewer 1 Report

The title, while cute, is misleading and should be more in line with the actual research.

You use "patient arbitrariness" when you are really describing patient non-adherence, which is not the same thing.  I would use nonadherence as a more accurate term.

Can you convert income to Euros (or US$)? Very few readers have any idea  what PLN 1000 means.

Reviewer 2 Report

Thank you for conducting the literature review and survey. This research question is a very urgent current situation globally. I expect this research will be disseminated widely after accepted.

Literature review of this paper discussed many areas/components of the with lots of good references. However, it feels like there are some tiny gaps in connecting the literature review and hypothesis of survey.

To avoid misunderstanding, please, make the clear that at the end of section 2 literature review. As well above the aim of study is to introduce concepts of patient satisfaction and patient arbitrariness as mediators in shaping overconsumption of health care. So you should explain more in detail how to connect variables and concept definition among the result literature review, hypothesis, the each concepts appeared on Figure 1 and variables on Table 1. It also should be discussed more.

At the end of the revise, please think of title to be more suitable, so that it appears consistent and the objective of the research.

I hope the research will help future cultural innovation in your country.

Reviewer 3 Report

Summary:
In the paper with the title "Is Dr House right? Do all patient lie? Understanding overconsumption in health systems", the authors
investigated based on a "Computer Assisted Telephone Interview" (CATI) factors of people in Poland that might affect/cause the overconsumption 
in health systems. The authors motivate their work, including an in-depth discussion on related works. Based on the latter, six hypotheses
are derived. Based on study data of a CATI in Poland, the authors performed descriptive statistics as well as a structural equation modelling 
approach to investigate the hypotheses. Results to these investigations are presented and discussed, including a discussion on limitations.
The authors conclude their work with four contributions and an outlook.

Points in favor:
- The paper is written very well
- The paper fits very well to the scope of the journal
- The paper deals with a topical subject, being interesting for the
  readers of the journal
- The paper draws a clear contribution
- The paper discusses related works properly
- The paper shows experimental results
- The paper discusses limitations explicitly

Points against the paper:
Although I like the paper and its results, some aspect should be improved:
- Related works are very extensively discussed, therefore they should be better summarized,
  in the best case by using a table
- The aspect of "lying" should be better motivated in the introduction, it is that prominently in the
  title, but only mentioned in the conclusion
- Limitations should be discussed within the discussion section
- In addition, the main contributions should be also put into the discussion
  -> This way the conclusion can be used just as a short conclusion
- Some minor language issues should be fixed, e.g.,
  Line 91: to to -> to
  Line 415: Another limitation is that related -> Another limitation is related
  Line 251: 3.2: indicated in Table -> indicated in Table 1
  I would recommend to write Table with a number always with a capital letter
- Please indicate significance levels
- Please use explaining legends in all tables, e.g.,
  Table 2: Although PLN is clear, please indicate its meaning in a legend
  Table 3: Meaning of values except Mean and SD
  Table 2: Instead of no. -> N 
- Use a picture for the SEM with the results
- Provide more background information to the CATI
- Which tools were used for the analyses

Round 2

Reviewer 3 Report

My concerns have been addressed, only thing that should be considered before publication, please check the writing of

nonadherence

I think it should be non-adherence